# Integrated Analysis of Metabolome and Transcriptome Reveals the Effect of Burdock Fructooligosaccharide on the Quality of Chinese Cabbage (*Brassica rapa* L. ssp. *Pekinensis*)

**DOI:** 10.3390/ijms252111459

**Published:** 2024-10-25

**Authors:** Xin Fu, Lixia Wang, Chenwen Liu, Yuxiang Liu, Xiaolong Li, Tiantian Yao, Jian Jiao, Rui Shu, Jingjuan Li, Yihui Zhang, Fengde Wang, Jianwei Gao

**Affiliations:** 1Shandong Key Laboratory of Bulk Open-Field Vegetable Breeding, Ministry of Agriculture and Rural Affairs Key Laboratory of Huang Huai Protected Horticulture Engineering, Institute of Vegetables, Shandong Academy of Agricultural Sciences, Jinan 250100, China; 15621321275@163.com (X.F.); 13148310672@163.com (L.W.); lcw123001@163.com (C.L.); nkylixiaolong@163.com (X.L.); yttme2013@163.com (T.Y.); darovejiji@163.com (J.J.); shrhero@163.com (R.S.); lijj0620@163.com (J.L.); zyh_0923@163.com (Y.Z.); 2School of Biological Science and Technology, University of Jinan, Jinan 250100, China; 15266165790@163.com

**Keywords:** burdock fructooligosaccharide, Chinese cabbage, metabolome, transcriptome, quality

## Abstract

Burdock fructooligosaccharide (BFO) is fructose with a low polymerization degree, which could improve the immunity to pathogens, quality, and stress resistance of vegetables. Still, there are no studies on applying BFO in Chinese cabbage. In this study, the effects of exogenous BFO sprayed with different concentrations (0, 5, 10, 20, 30 g·L^−1^) on the growth and soluble sugar content of Chinese cabbage seedlings were determined. The result showed that 10 g·L^−1^ was the appropriate spraying concentration. Based on metabolome analysis, a total of 220 differentially accumulated metabolites (DAMs) were found, among which flavonoid metabolites, glucosinolate metabolites, and soluble sugar-related metabolites were the key metabolites involved in improving the quality of Chinese cabbage caused by BFO. Further combination analysis with transcriptome, trans-cinnamate 4-monooxygenase (*CYP73A5*), and chalcone synthase 1 (*CHS1*) were more closely associated with the DAMs of flavonoid biosynthesis. Sulfotransferases 18 (*SOT18*), Branched-chain amino acid amino transferases 6 (*BCAT6*), and cytochrome P450 monooxygenase (*CYP83A1*) were the key genes in glucosinolate biosynthesis. Hexokinase (*HxK1*), beta-glucosidase 8 (*BGL08*), invertase 3 (*INV3*), beta-glucosidase 3B (*BGL3B*), and sucrose phosphate synthase 1 (*SPS1*) were significantly upregulated, potentially playing crucial roles in the soluble sugar metabolism. In conclusion, these results provided an understanding of the effects of BFO on the expression of genes and the accumulation of metabolites related to quality formation in Chinese cabbage.

## 1. Introduction

Chinese cabbage (*Brassica rapa* L. ssp. *Pekinensis*) originated in China, and is the vegetable crop with the largest planting area and the highest yield in China. It is also one of the most important vegetable crops worldwide. Chinese cabbage is rich in nutrients such as crude fiber, sugar, vitamins, and essential trace elements [1,2]. In addition, flavonoids and glucosinolates are the main components that affect the nutritional quality, flavor quality, and appearance quality of Chinese cabbage [3,4]. In recent years, with the improvement of people’s living standards, Chinese cabbage with good taste and high nutritional value is more and more favored by consumers, and the quality of Chinese cabbage has become the focus of research.

Burdock (*Arctium lappa* L.) is a biennial herb of the genus Arctium in the composite family. Burdock root contains a high amount of health-beneficial antioxidant compounds such as caffeine, flavonoids, and lignans [5], which can be eaten as a vegetable and also have high medicinal value. Burdock fructooligosaccharide (BFO) is a high-purity fructose extracted from burdock root with a molecular weight of less than 3KD and is an inulin oligosaccharide with 13 degrees of polymerization [6]. Many studies have shown that oligosaccharides can be used as signaling molecules to participate in biological processes such as regulating plant growth and development and activating various defensive reactions in plants, thereby improving crop resistance and yield [7,8,9,10]. However, most earlier studies were related to chitosan and chitooligosaccharides, which have high production costs and poor stability [11]. It is worth noting that the extraction process of BFO is simple, the production cost is low, and the extraction rate can reach more than 85%. Consequently, it can be widely used [6].

As a signal, BFO can induce stomatal closure mediated by reactive oxygen species (ROS) and ROS-dependent nitric oxide, upregulate the related genes of hormone signal transduction and secondary metabolite synthesis, and improve immunity to defense [12,13]. As an inducer for postharvest disease control, BFO can activate the salicylic acid-dependent signaling pathway and inhibit postharvest browning of Kyoho grapes [14] as well as BFO can induce the explosion of hydrogen peroxide and the activity of antioxidant enzymes to inhibit the invasion of *A. tenuissima* and the *Alternaria* fruit rot in postharvest blueberries [15]. In addition, BFO inhibits sucrose output, directly retaining more carbohydrates to improve fruit sweetness and indirectly maintaining cell osmotic pressure and hardness in the mesocarp. BFO inhibits the entry of sucrose, thereby slowing down the rot process of grapes in the exocarp [16]. Exogenous application of BFO can improve the plant’s stress resistance and enhance the quality of vegetables [17]. Applying foliar fertilizer to watermelon with BFO as the main component increased the content of vitamin C and sugar in watermelon while significantly increasing the watermelon yield [18]. Dong et al. [19] stated that spraying BFO on spinach seedlings increased the total sugar content and reduced the nitrate and oxalate content, thus improving the quality of spinach. Furthermore, BFO can enhance the photochemical efficiency and the resistance of cucumber seedlings under cold stress, while the appearance quality and nutritional quality are significantly advanced [20,21].

Although the role of BFO in enhancing plant defense against pathogens, improving postharvest preservation, and regulating growth and quality have been extensively researched, the effects of BFO on the growth and quality of Chinese cabbage have not been reported, the regulatory mechanism of BFO on the quality of Chinese cabbage is still unclear. At present, with the development of omics technology, comprehensive analysis of transcriptome and metabolome provides a new understanding of the interaction between genes and metabolites in plants, which is helpful in further identifying target genes and research directions. In this study, different concentrations of BFO were used to treat the Chinese cabbage seedlings, and the growth and the soluble sugar content were measured to determine the optimal application concentration. By analyzing the transcriptome and metabolome, the effects of BFO on the transcriptional and metabolic levels of Chinese cabbage seedlings were explored, and the differential genes and metabolites affected by BFO were excavated. This study will provide a theoretical basis for revealing the effect of BFO on the accumulation of key components in Chinese cabbage and lay a foundation for further revealing the molecular mechanism of BFO regulation of Chinese cabbage quality.

## 2. Result

### 2.1. Effects of Different Concentrations of Burdock Fructooligosaccharide (BFO) on the Growth and Quality of Chinese Cabbage Seedlings

BFO plays a vital role in improving plant resistance and quality. In Figure 1A, exogenous BFO with different concentrations influenced the fresh weight of Chinese cabbage seedlings. Compared with 0 g·L^−1^, the 5 g·L^−1^ treatment showed no significant difference. The 10 g·L^−1^ and 20 g·L^−1^ BFO treatments increased the fresh weight of seedlings by 28.60% and 23.45%, respectively. However, the 30 g·L^−1^ BFO treatment did not differ significantly from the 0 g·L^−1^. Meanwhile, the results showed that with the increase in BFO concentration, the contents of soluble sugar showed a trend of first increasing and then decreasing. In comparison with 0 g·L^−1^, soluble sugar content increased by 17.42%, 36.36%, 31.82%, and 25.00%, respectively (Figure 1B), under BFO concentrations of 5, 10, 20, and 30 g·L^−1^. Therefore, a BFO concentration of 10 g·L^−1^ was the optimal concentration to promote the growth and quality of Chinese cabbage. In addition, we applied 10 g·L^−1^ BFO to two fast-growing vegetable varieties, ‘863’ (T1) and ‘Champion’ (T2), and the results showed that BFO could also significantly increase the fresh weight of T1 and the soluble sugar content of T1 and T2 (Appendix A).

In addition, to explore the time-lasting effect of exogenous BFO on endogenous sugar accumulation of Chinese cabbage seedlings, the changes in soluble sugar content in leaves were measured at 0–5 d after 10 g·L^−1^ BFO treatment. The results showed that the content of soluble sugar gradually increased with sample time and reached the highest level at 3 d, which increased by 49.56% compared with 0 d, and then gradually decreased (Figure 1C), indicating that the content of endogenous sugar was the highest after BFO treatment for 3 d.

### 2.2. Metabolome Analysis of BFO on Chinese Cabbage Seedlings

To explore the effect of BFO on the metabolic level of Chinese cabbage seedlings, exogenous 10 g·L^−1^ BFO treated for 3 d were sampled to determine the non-targeted metabolome, with the H_2_O treatment as the control (CK). The Orthogonal Partial Least Squares-Discriminant Analysis (OPLS-DA) diagram showed that there was a big difference between the BFO and CK in the horizontal coordinate. R2X and R2Y indicate that the interpretation rate of the built model for the X and Y matrix is 0.82 and 1, respectively, and Q2Y demonstrates that the prediction ability of the model is 0.995, which indicates that the data of this metabolome are reliable, and there is a significant difference between the BFO and CK treatment (Figure 2A). A total of 1651 substances were detected in the metabolome, and 220 differentially accumulated metabolites (DAMs) were found, with variable importance in projection (VIP) > 1 and fold change (FC) > 2 as thresholds (Appendix A). Among them, 98 metabolites were upregulated, including flavonoids, benzene and substituted derivatives, heterocyclic compounds, and other substances. There are 122 downregulated metabolites, mainly including benzene and substituted derivatives, heterocyclic compounds, amino acids, and derivatives (Figure 2B,D). Kyoto Encyclopedia of Genes and Genomes (KEGG) analysis of DAMs found that they were mainly enriched in flavonoid biosynthesis, secondary metabolite biosynthesis, ubiquinone biosynthesis, fructose and mannose metabolism, and glucosinolate biosynthesis (Figure 2C). These results suggest that BFO may affect the quality-related metabolites of Chinese cabbage seedlings.

### 2.3. Transcriptome Analysis of BFO on Chinese Cabbage Seedlings

The transcriptomic analysis was performed to investigate the effect of BFO on transcriptional expression levels in Chinese cabbage. The results showed that each sample group obtained over 6.7 G clean bases. The percentage of bases with Phred values exceeding 30 in total bases (Q30) was greater than 96%, and the comparison rate with the reference genome (Brara_Chiifu_V3.0) was greater than 84%, indicating that the transcriptome sequencing quality was good and could satisfy the needs of follow-up analysis (Table 1).

Principal component analysis (PCA) of the transcriptome showed that PC1 and PC2 separated BFO from CK. The contribution rate of PC1 was 45.69%, and that of PC2 was 30.56%, indicating that BFO could significantly affect the transcriptional level of Chinese cabbage seedlings (Figure 3A). There are 2249 differentially expressed genes (DEGs) induced by BFO treatment compared with CK, including 1328 up-regulated genes and 921 down-regulated genes (Figure 3B; Appendix A). The DEGs were subjected to GO functional enrichment analysis and were divided into three categories: molecular function (MF), biological process (BP), and cellular component (CC). The MF category contains the most DEGs, and the significantly enriched MF terms were “oxidoreductase activity”, “heme binding”, and “tetrapyrrole binding” terms. In the BP category, “cell wall organization or biogenesis” had the most DEGs. In the CC category, the “extracellular region” was significantly enriched (Figure 3C).

Additionally, KEGG enrichment analysis result found that the DEGs were significantly enriched in glucosinolate biosynthesis and phenylpropanoid biosynthesis. The analysis also showed higher counts of DEGs in “carbon metabolism”, “biosynthesis of amino acids”, and “starch and sucrose metabolism” (Figure 3D). These results suggested that BFO could affect the expression of genes related to growth and quality of Chinese cabbage seedlings.

### 2.4. Combined Metabolome and Transcriptome Analysis of BFO on Chinese Cabbage Seedlings

To further explore the effect of BFO on Chinese cabbage, we performed an integrated analysis of transcriptome and metabolome based on the KEGG metabolic pathway (Appendix A). Through the analysis of the number of DAMs and DEGs in different metabolic pathways, we found that they were more enriched in the “Biosynthesis of cofactor”, “2-Oxocarboxylic acid metabolism”, “Biosynthesis of amino acid” pathway (Figure 4A). Analysis of the significance of the difference (*p*-value < 0.05) showed that both DAMs and DEGs were significantly enriched in “Flavonoid biosynthesis” and “Ubiquinone and another terpenoid-quinone biosynthesis”. Furthermore, one “Glucosinolate biosynthesis” was particularly interesting, as DEGs differed most significantly among all KEGG pathways (Figure 4B). To analyze the relationship between DAMs and DEGs of Chinese cabbage seedlings in response to BFO, the correlation analysis of “Biosynthesis of cofactor”, “2-Oxocarboxylic acid metabolism”, “Biosynthesis of amino acid”, “Ubiquinone and another terpenoid-quinone biosynthesis”, “Flavonoid biosynthesis” and “Glucosinolate biosynthesis” pathway were conducted and made the clustering heat map with the Pearson correlation coefficient greater than 1 (Figure 4C; Appendix A). The result showed that BFO may regulate a complex network related to growth and quality, controlling the metabolic balance of cofactors and promoting the accumulation of different nutrients, thereby regulating the growth and quality of Chinese cabbage.

The flavonoid biosynthesis pathway has shown significant differences with enriched DEGs and DAMs. To gain a deeper understanding of the effect of BFO on the flavonoid biosynthesis metabolic pathway, the flavonoid biosynthesis pathway map was constructed, and DEGs and DAMs were labeled (Figure 5A). There are five DAMs in the flavonoid biosynthesis pathway, wherein BFO could promote the accumulation of kaempferol, quercetin, (-)-epicatechin, (+)-gallocatechin, and decrease of desmethylxanthohumol. Kaempferol and quercetin are flavonols, (-)-epicatechin and (+)-Gallocatechin are flavanols, which can further participate in anthocyanins biosynthesis. According to the transcriptome analysis, the gene expression of trans-cinnamate 4-monooxygenase (*CYP73A*), caffeoyl-CoA O-methyltransferase (*CAMT*), chalcone synthase (*CHS*), chalcone isomerase (*CFI*), anthocyanidin synthase (*ANS*) and flavonol synthase (*FLS*) were significantly upregulated (Figure 5B). Notably, CYP73A, a key enzyme in the phenylpropanoid pathway, belongs to the plant cytochrome P450 monooxygenase family. CHS, derived from the polyketone synthase III family, is a rate-limiting enzyme in flavonoid biosynthesis. Flavonols, the most widely distributed and abundant flavonoid substances, are catalyzed by FLS, a key enzyme in flavonol biosynthesis. DEGs and DAMs correlation analysis with the Pearson correlation coefficient greater than 0.8 revealed that the down-regulated metabolite desmethyl-xanthohumol was significantly negatively correlated with all DEGs. The key genes of flavonoid biosynthesis *BraA03gCYP73A5* and *BraA10gCHS1* were more associated with DAMs (Figure 5C,D; Appendix A).

Glucosinolate is an essential secondary metabolite in brassicaceous vegetables. Aliphatic glucosides can be synthesized using methionine as a precursor, and methylthio-glucosinolate, including glucoerucin and glucolequerellin, accumulated significantly under BFO treatment in the metabolome. Aromatic amino acids such as phenylalanine are precursors for synthesizing aromatic glucosinolate. 2-phenylethyl-glucosinolate of gluconasturtiin accumulated significantly, but benzyl-glucosinolate of glucotropeaolin decreased under BFO treatment (Figure 6A). Glucosinolate biosynthesis mainly involves the elongation of side chains, formation of core structures, and modification of side chains. Branched-chain amino acid aminotransferases (*BCAT*) and methylthioalkylmalate synthases (*MAM*) play important roles in the process of side-chain elongation. In this process, *BraA03gBCAT4*, *BraA05gBCAT4,* and *BraA02gMAM1* were significantly up-regulated, while *BraA08gBCAT2* and *BraA06gBCAT6* were down-regulated. In the glucosinolate core pathway, long-chain methionine was oxidized by cytochrome P450 monooxygenase (*CYP79F1*, *CYP83A1*) to aldoxime and conjugated to form s-alkyl thiooxime, then split by C-S lyase (*SUR1*) to thiooxime acid and transformed to desulfo-glucosinolate by UDP-glucosyl transferase (*UGT74*). Finally, sulfotransferases (*SOT*) catalyzed desulfurization to form glucosinolate (Figure 6B). DEGs and DAMs correlation analysis found that glucotropeaolin has a positive correlation with the downregulated DEGs. Additionally, *BraA07gSOT18*, *BraA06gBCAT6,* and *BraA04gCYP83A1* were more closely associated with glucosinolate biosynthesis (Figure 6C,D; Appendix A).

BFO significantly increased the soluble sugar content in Chinese cabbage seedlings (Figure 1B). To further study the effects of BFO on soluble sugar biosynthesis, we analyzed the soluble sugar metabolism pathways in both transcriptomes and metabolites. According to the metabolome analysis, BFO can significantly induce the accumulation of D-fructose, Sucrose 6’-phosphate, and GDP-D-mannose (Figure 7A). In the starch and sucrose metabolism, as well as the fructose and mannose metabolism, the expression of invertase (*INV*) and hexokinase (*HxK1*) was significantly upregulated. Furthermore, sucrose metabolism-related enzymes of sucrose phosphate synthase (*SPS*), sucrose-6-phosphatase (*SPP*), and beta-glucosidase (*BGL/BGH*) showed both up-regulated and down-regulated genes (Figure 7B). Among these, *BraA06gSPS2*, *BraA04gSPP2*, *BraA05gBGL15*, and *BraA04gBGL13* were significantly down-regulated under BFO treatment. Correlation analysis indicated that these genes were negatively correlated with D-fructose, Sucrose 6′-phosphate, and GDP-D-mannose. Based on the correlation analysis results, *BraA10gHxK1*, *BraA08gBGL40*, *BraA06gINV3*, *BraA03gBGL3B,* and *BraA02gSPS1* appear to play important roles in the BFO affected soluble sugar metabolism pathway (Figure 7C,D; Appendix A).

### 2.5. Validation of the DEGs by qRT-PCR

To verify the reliability of transcriptome data, 12 DEGs were randomly selected to identify their relative expression levels under CK and BFO treatment. qRT-PCR results showed that the expression patterns of genes related to flavonoid synthesis(*BraA03gCYP73A5*, *BraA01gANS*, *BraA10gFLS1*), glucosinolate biosynthesis (*BraA04gCYP83A1*, *BraA04gCYP83A2*, *BraA02gMAM1*, *BraA07gSOT18*), soluble sugar synthesis (*BraA02gSPS1*, *BraA08gSPP1*, *BraA08gBGL40*, *BraA06gINV3*) and ubiquinone and another terpenoid-quinone biosynthesis (*BraA05g4CL2*) were consistent with the FPKM values form RNA-seq. Specifically, these genes were significantly upregulated under BFO treatment (Figure 8). The above results indicate that the transcriptome data in this study were accurate and reproducible, demonstrating that BFO could improve the quality of Chinese cabbage.

## 3. Discussion

BFO, a fructooligosaccharide extracted and purified from burdock root, is a functional fructose with important function medical benefits such as diabetes prevention and cellular immunity [22,23,24] and also plays a vital role in plants [25]. Previous studies have found that BFO could induce systemic resistance in plants, improve the resistance to bacterial infection [12], and alleviate the process of fruit corrosion by bacteria after harvest [15]. For fruit and vegetable quality, exogenous BFO application could promote the growth of cucumber plants [21] and enhance the nutritional quality and flavor quality of watermelon and spinach [18,19]. In this study, we explored the effects of different concentrations of exogenous BFO on the growth and quality of Chinese cabbage seedlings. We found that 10 g·L^−1^ BFO significantly increased the Chinese cabbage seedlings’ fresh weight and soluble sugar content. However, the induction effect decreased as the enhancement of BFO concentration. Therefore, 10 g·L^−1^ BFO was the optimal concentration for exogenous application on Chinese cabbage seedlings. In addition, 10 g·L^−1^ BFO treated the seedlings reached the maximum value of soluble sugar at 3 d (Figure 1). Transcriptomics is an essential method for exploring genome function and differential expression. It plays an important role in studying plant growth, development, and stress resistance [26]. On the other hand, metabolomics is an analytical technique that can directly reflect the changes and differences in the metabolic level and then speculate related metabolic pathways and networks [27]. Plants’ responses to various environments will change gene expression and accumulate metabolites, and multi-omics analysis has been widely used to reveal these biological processes [28,29]. In the present study, we performed metabolome and transcriptome analyses on Chinese cabbage seedlings treated with 10 g·L^−1^ BFO for 3 d, and the results revealed 220 DAMs and 2249 DEGs. KEGG analysis of DAMs was mainly enriched in flavonoid biosynthesis, secondary metabolite biosynthesis, ubiquinone biosynthesis, fructose and mannose metabolism, and glucosinolate biosynthesis. Additionally, KEGG analysis of DEGs was significantly enriched in glucosinolate biosynthesis and phenylpropanoid biosynthesis (Figure 2 and Figure 3), indicating that BFO could affect Chinese cabbage seedlings’ metabolic level and transcription level. Flavonoids, glucosinolates, soluble sugars significant, and other related metabolic substances and genes play major roles in jointly regulating and improving the nutritional quality of Chinese cabbage.

Flavonoids are secondary metabolites in plants, including seven subclasses of anthocyanin, proanthocyanidins, flavonols, flavonoids, flavanones, isoflavones, and chalcones [30], which not only play a vital role in plant resistance to environmental stress but also is the primary source of plant pigment [31,32]. Furthermore, flavonoids exhibit various beneficial effects, including free radical scavenging, anti-oxidation, regulation of glucose and lipid metabolism, and enhancement of body immunity [33]. The flavonoid biosynthesis pathway was significantly enriched and has many DAMs and DEGs by the integrated analysis of transcriptome and metabolome (Figure 4). This plant pathway begins with phenylpropanoid biosynthesis, and p-coumaroyl CoA is catalyzed by C4H. C4H, a key enzyme in the phenylpropanoid biosynthesis, belongs to the cytochrome P450, derived from the CYP73 family [34,35]. Then, naringenin chalcone is produced under CHS, and dihydroflavonol is catalyzed by CHI and F3H. CHS is a rate-limiting enzyme in flavonoid biosynthesis [36], while CHI can accelerate chalcone’s isomerization and greatly increase the rate of catalytic reactions [37]. Similarly, caffeoyl-CoA and feruloyl-CoA are converted to dihydroquercetin and dihydromyricetin, respectively. FLS catalyzes the desaturation of dihydroflavonol to form flavonols. Kaempferol, quercetin, and (+)-Gallocatechin are formed from dihydrokaempferol, dihydroquercetin, and dihydromyricetin as substrates catalyzed by FLS. In addition, leucoanthocyanidin can be generated by dihydroflavonols under the action of DFR, and the (-)-Epicatechin is catalyzed by ANS [38,39]. As shown in Figure 5, flavonoids such as kephenol, quercetin, (-)-epicatechin and (+)-gallatechin significantly accumulated under BFO treatment. Concurrently, the expressions of flavonoid synthesis pathway genes *CYP73A*, *CAMT*, *CHS*, *CFI*, *ANS,* and *FLS* were up-regulated considerably; these results indicated that BFO could accumulate flavonoids by regulating flavonoid pathway and improve the nutritional quality of Chinese cabbage seedlings. Moreover, the correlation analysis of DAMs and DEGs in the flavonoid pathway found that *BraA03gCYP73A5* and *BraA10gCHS1*, as key rate-limiting enzymes in phenylpropanoid metabolism and flavonoid biosynthesis, were most correlated with flavonoids, respectively (Figure 5), demonstrating *BraA03gCYP73A5* and *BraA10gCHS1* play an important role in the flavonoid biosynthesis of Chinese cabbage.

Glucosinolate is an important secondary metabolic substance of cruciferous vegetables, which plays an important role in plant resistance to biotic and abiotic stresses, improvement of flavor quality and nutritional quality, and prevention of cancer [40,41,42,43]. According to the different amino acids of R side chain groups, it can be divided into three categories, including aliphatic glucosinolate with a side chain derived from methionine, indole glucosinolate with a side chain derived from tryptophan, and aromatic glucosinolate with a side chain derived from phenylalanine [44]. The biosynthesis of amino acids could affect the glucosinolate biosynthesis pathway. In this study, the glucosinolate biosynthesis pathway and related amino acid biosynthesis pathway enriched more DAMs and DEGs in the transcriptome and metabolome (Figure 4). The aliphatic glucosinolate of glucoerucin and glucolequerellin, as well as the aromatic glucosinolate of gluconasturtiin, was accumulated significantly, while the content of glucotropeaolin was decreased in response to BFO treatment (Figure 6A). *BCAT* plays a role in deamination and transamination of aliphatic glucoside side chain elongation [45,46], in which *BCAT4* is involved in the first step of side chain elongation and also involved in the regulatory reaction of plant defense [47]. The *MAM* gene family mainly controls the chain elongation of aliphatic glucoside and has different functions in different ecotypes of plants [48]. The core structure of glucosinolate is composed of β-D-thioglucoyl and sulfonic oxime groups, and the *CYP79* gene family, *CYP83* gene family, *SUR1*, *UGT74B1* and *SOT* gene family participate in the glucosinolate core pathway, catalyzing the synthesis both of aliphatic glucosinolate and aromatic glucosinolate from the chain elongation amino acids [49,50]. Among the *SOTs*, *SOT16* is responsible for desulfurizing long-chain aromatic glucosinolate, while *SOT17* and *SOT18* are responsible for desulfurizing methionine-derived glucosinolate [51]. In our study, the expression of *BACT4*, *MAM1*, *CYP79A1*, *CYP83A1*, *CYP83A2*, *CYP83B1*, *CYP83B3*, *SUR1*, *UGT47B1* and *SOT16*, *SOT17*, *SOT18* were significantly induced by the BFO treatment. The correlation analysis found that *BraA07gSOT18*, *BraA06gBCAT6*, and *BraA04gCYP83A1* were more associated with the DAMs of glucosinolate biosynthesis (Figure 6B–D); we proposed that the *BraA07gSOT18*, *BraA06gBCAT6* and *BraA04gCYP83A1* might be the essential gene of BFO regulating glucosinolates accumulation in Chinese cabbage seedlings.

The soluble sugar content is a crucial factor determining Chinese cabbage’s taste quality. BFO is a kind of fructose, and exogenous BFO can significantly increase the soluble sugar content in Chinese cabbage seedlings (Figure 1B). Therefore, it is of great significance to investigate the metabolism and transcription levels of soluble sugar accumulation induced by BFO in Chinese cabbage to improve its quality. Soluble sugars mainly include fructose, glucose, and sucrose, which are important factors for determining sweet taste. Fructose has the highest sweetness [52]. In the metabolome of this study, D-fructose, Sucrose 6’-phosphate, and GDP-D-mannose accumulated under BFO treatment, significantly increasing the soluble sugar content of Chinese cabbage (Figure 7A). This is consistent with previous studies that show that BFO can alleviate the reduction of soluble sugar and maintain postharvest fruit sweetness [16]. Moreover, GDP-D-mannose is a molecule that serves as a source of mannosyl units in various glycoconjugates, which play a crucial role in biochemical processes such as the synthesis of deoxy sugars and L-ascorbic acid in plants [53]. This means that the increase in mannose content may contribute to improving the nutritional quality of Chinese cabbage. The regulatory network of soluble sugar metabolism is complex.

Genes encoding various enzymes play crucial roles in this metabolism process. Invertase (INV) catalyzes cytoplasmic sucrose to produce D-Fructose and D-Glucose-6P, and D-Fructose produces D-Fructose 6-phosphate and D-Mannose-6-phosphate in response to HXKs. Sucrose can also produce UDP-glucose (UDPG) with sucrose synthase (SUS) catalysis, and UDPG can synthesize sucrose again with SPS and SPP catalysis [54]. In addition, *BGL/BGH* belongs to the GH1 gene family and can completely hydrolyze UDPG to D-Glucose [55], which has the function of cellulose degradation [56]. In this study, genes encoding INV and HxK1 were significantly upregulated, and the genes encoding SPS, SPP, and BGL/BGH were both up- and down-regulated under the BFO treatment (Figure 7A,B). the above DEGs collectively regulate the soluble sugar metabolism pathway in Chinese cabbage. Among them, *BraA10gHxK1*, *BraA08gBGL40*, *BraA08gINV3*, *BraA03gBGL3B,* and *BraA02gSPS1* may be closely related to the accumulation of soluble sugars induced by BFO with the correlation analysis.

Due to BFO’s effectiveness in improving the quality of Chinese cabbage, it can be promoted as an exogenous fertilizer and has a good development and utilization value in application. To further apply to production, the next research can consider the effect of BFO on the growth and quality of Chinese cabbage during the whole growth period, the general applicability of BFO to vegetable crops as well as the advantages of BFO compared with other chitosan or foliar fertilizers. In short, BFO has broad prospects in the field of physiological function and practical application research.

## 4. Materials and Methods

### 4.1. BFO Preparation

BFO was prepared according to the method of Wang et al. [57].

### 4.2. Plant Materials and Treatment

‘Gaolv No. 1’ was used as the experiment variety. The full and uniform seeds were selected and planted in a pot (7 cm × 7 cm) filled with substrate:vermiculite = 1:1, then placed in an artificial culture room with a temperature of 20 ± 2°C, photoperiod of 16 h/8 h (day/night), photon flux density (PFD) of 100 μmol·m^−2^·s^−1^, and relative humidity of 30–50%.

To screen the most appropriate concentration of burdock oligosaccharides (BFO), seedlings of similar size were chosen after seeding for 25 days. The concentrations of 0 g·L^−1^, 5 g·L^−1^, 10 g·L^−1^, 20 g·L^−1^ and 30 g·L^−1^ BFO solution with 0.01% Tween-20 were prepared, respectively. Using ten plants per treatment, and the leaves of Chinese cabbage seedlings were sprayed once every 5 days for a total of three times. After first being treated for 15 days, the leaves were sampled to determine the fresh weight and the soluble sugar content.

To explore the temporal effects of BFO, we sampled the treated seedlings at 0, 1, 2, 3, 4, and 5 days and determined their soluble sugar content. For transcriptome and metabolome sequencing, the 10 g·L^−1^ BFO treated for 3 d seedlings was sampled, and the H_2_O treatment was used as the control.

### 4.3. Fresh Weight Measurement

Cut off the plant close to the soil and wipe off the soil on the surface, using a microbalance (Sartorius, Beijing, China) to determine its weight as fresh weight.

### 4.4. Soluble Sugar Content Measurement

Soluble sugar content was determined by the anthranone method [58]. A total of 2 g Chinese cabbage sample was extracted from 10 mL distilled water. After boiling for 30 min, filter into a volumetric bottle and set the volume to 100 mL. Then, add 0.5 mL extract solution, 1.5 mL distilled water, 0.5 mL anthrone-ethyl acetate reagent, and 5 mL concentrated sulfuric acid into the test tube, respectively. Mix and boil the test tube for 1 min. The absorbance was measured at the wavelength of 630 nm.

### 4.5. Metabolome

A total of 120 mg ground samples were extracted with 1 mL solution of methanol:acetonitrile:water = 2:2:1, 200 W ultrasonic for 10 min and centrifuge 15,871× *g* for 5 min (Eppendorf, Hamburg, Germany), the supernatants were concentrated and dried with a low-temperature vacuum concentrator add 100ul methanol and then dissolve, filter with 0.22 μm filter and waiting for the HPLC-MS analysis.

Chromatographic mass spectrometry parameters were set as follows:

Thermo HYPERSIL GOLD aQ C18 Chromatographic column (2.1 × 100, 1.9 μm); Column temperature of 3 °C; Mobile phase of aqueous solution containing 0.1% acetic acid and acetonitrile solution containing 0.1% acetic acid; Flow rate of 0.3 mL·min^−1^; Sample size of 3 μL.

MS positive ion mode: Spray voltage of 3.8 kv; Sheath gas of 40 Arb; Auxiliary gas of 10 Arb; Ion transfer tube temperature of 350 °C.

MS negative ion mode: Spray voltage of 2.9 kv; Sheath gas of 40 Arb; Auxiliary gas Arb 0 Arb; Ion transfer tube temperature of 350 °C.

### 4.6. Transcriptome Analysis

The total sample of RNA was extracted, and the library was constructed after accurately detecting RNA integrity and total amount using an Agilent 2100 bioanalyzer (Agilent, Santa Clara, CA, USA). Illumina (Illumina, San Diego, CA, USA)sequencing was performed, and a paired-end reading of 150 bp was generated. Clean data were obtained by removing a small number of reads with sequencing connectors or with low sequencing quality from the original data. At the same time, the Q20, Q30, and GC contents of the clean data were calculated. Brara_Chiifu_V3.0 as the reference genome (http://www.brassicadb.cn/#/SearchJBrowse/?Genome=Brara_Chiifu_V3.0, accessed on 29 October 2020) compared with the paired-end clean read by HISAT2 v2.0.5. Using *p* value < 0.05 and |log2foldchange| > 2 as the threshold of significant differential expression, ClusterProfiler software (3.8.1) was used to perform GO enrichment analysis and statistical enrichment in Kyoto Encyclopedia of Genes and Genomes (KEGG) pathways of different-expressed gene.

### 4.7. Real-Time Quantitative PCR (RT-qPCR) Analysis

The TRNzol Universal (TIANGEN, Beijing, China) was used to extract the total RNA in Chinese cabbage leaves. The purity and quantity of RNA were verified using a Micro-Spectrophotometer (KAIAO, Beijing, China). Subsequently, RNA was used to synthesize the cDNA with StarScript III All-in-one RT Mix with gDNA Remover (GenStar, Beijing, China). The qPCR was conducted with 2×RealStar Fast SYBR qPCR Mix (GenStar, Beijing, China) with the gene-specific primers (Table 2). Calculation of the relative quantification was performed using the comparative 2^−ΔCT^ method.

### 4.8. Statistical Analysis

The whole experiment was performed at least in triplicate. All data were analyzed and plotted using Microsoft Office Excel version 2019. Using the DPS version 7.05 soft to proceed, difference significance analysis and the statistical analysis of the values were determined at *p* < 0.05, according to Duncan’s multiple range tests.

## 5. Conclusions

In this study, the effects of exogenous BFO on the growth and quality of Chinese cabbage seedlings were studied, and DAMs, as well as DEGs, were further analyzed by metabolomics and transcriptomics to explore the regulation mechanism of BFO regulation of Chinese cabbage quality. Results showed that 10 g·L-1 BFO was the optimal concentration for Chinese cabbage, regulating flavonoid biosynthesis, glucosinolate biosynthesis, and soluble sugar metabolism to improve the flavor quality and nutritional quality of Chinese cabbage. BFO could induce the accumulation of flavonols, which are more associated with the expression of *CYP73A5* and *CHS1*. *SOT18*, *BCAT6*, and *CYP83A1* might be the essential gene for the increase of methylthio-glucosinolate and 2-phenylethyl-glucosinolate. Furthermore, D-fructose, Sucrose 6′-phosphate, and GDP-D-mannose were significantly accumulated with the upregulation of *HxK1*, *BGL08*, *INV3*, *BGL3B,* and *SPS1*. In summary, this study revealed the mechanism of BFO regulating the expression of quality-related metabolites and genes, providing valuable insights for improving the quality of Chinese cabbage.

## Figures and Tables

**Figure 1 ijms-25-11459-f001:**
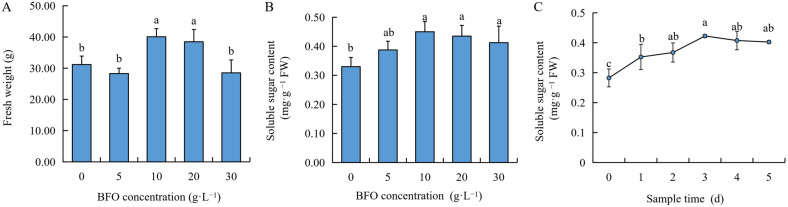
Effect of BFO treatment on seedling the growth and quality of Chinese cabbage. (**A**), Effect of BFO treatment on fresh weight in Chinese cabbage seedlings. (**B**), Effect of BFO treatment on the soluble sugar content of Chinese cabbage seedlings. (**C**), Changes of soluble sugar content in Chinese cabbage seedlings with different sampling days. Data are the means of three replicates (±SDs). Different letters indicate the significant differences between samples according to Duncan’s new multiple range test (*p* < 0.05).

**Figure 2 ijms-25-11459-f002:**
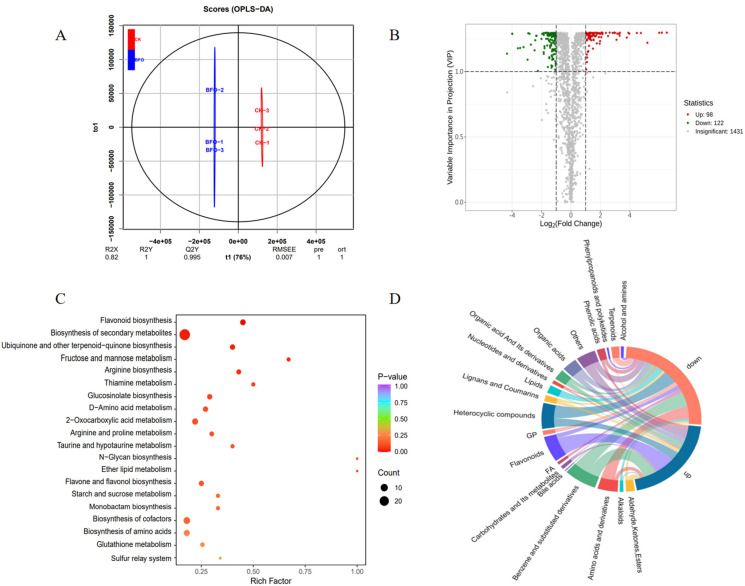
Metabolic profiling of Chinese cabbage in response to exogenous BFO. (**A**), OPLS-DA of metabolites identified by BFO and CK treated samples. (**B**), Volcano plots for DAMs between BFO and CK-treated samples. (**C**), KEGG pathway enrichment analysis of DAMs. (**D**), The categories of upregulated and downregulated DAMs.

**Figure 3 ijms-25-11459-f003:**
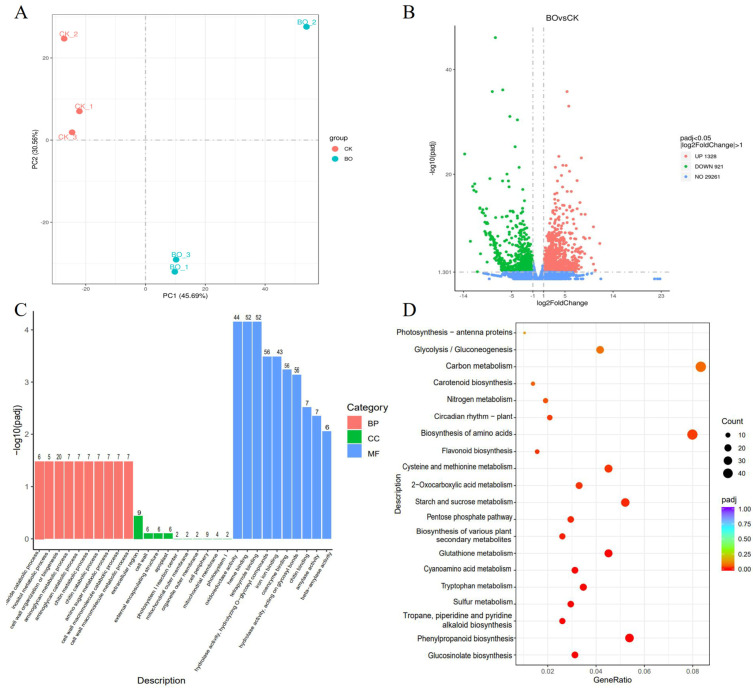
Transcriptional profiling of Chinese cabbage in response to exogenous BFO. (**A**), PCA analysis of DEGs identified by BFO and CK treated samples. (**B**), Volcano plots for DEGs between BFO and CK-treated samples. (**C**), GO enrichment analysis of DEGs. (**D**), KEGG pathway enrichment analysis of DEGs.

**Figure 4 ijms-25-11459-f004:**
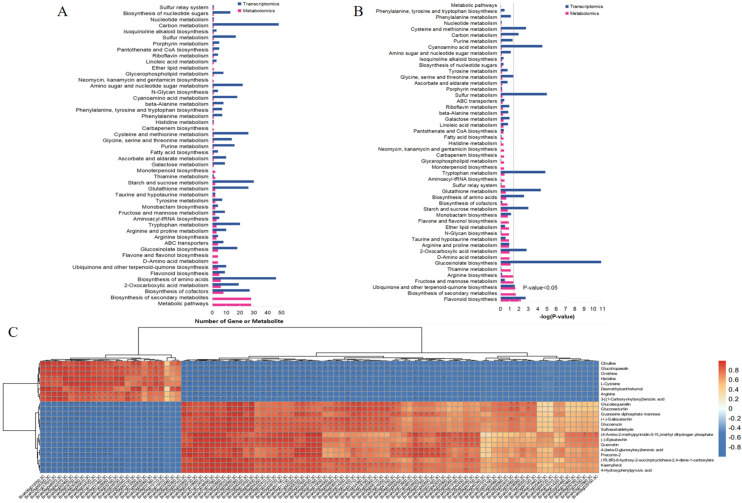
Combined metabolome and transcriptome analysis of Chinese cabbage in response to exogenous BFO. (**A**), The number analysis of DEGs and DAMs in Chinese cabbage in the KEGG pathway was identified by transcriptome and metabolome. (**B**), The *p*-value analysis of DEGs and DAMs in Chinese cabbage in the KEGG pathway identified by transcriptome and metabolome. (**C**), The correlation heatmap analysis of DEGs and DAMs in Chinese cabbage. Asterisks indicate statistical significance using pearson correlation analysis: * *p* ≤ 0.05, ** *p* ≤ 0.01,****p* ≤ 0.001.

**Figure 5 ijms-25-11459-f005:**
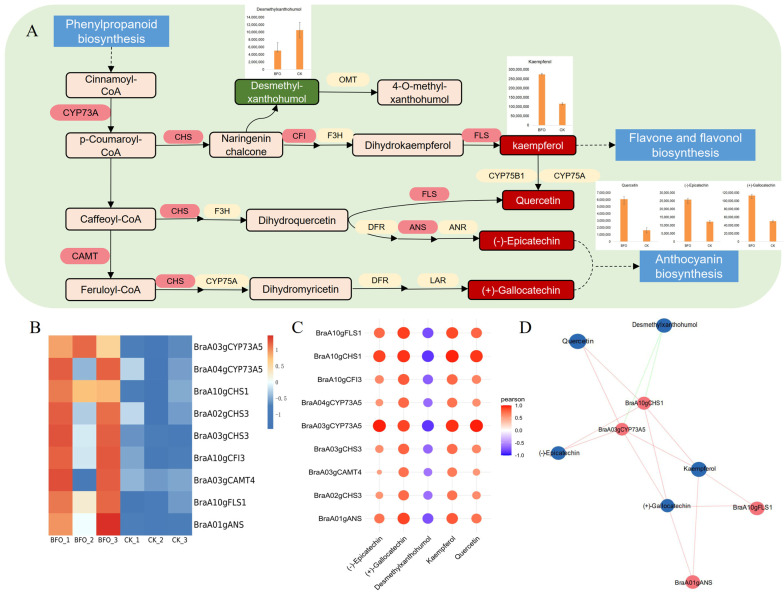
The DEGs and DAMs involved in flavonoid biosynthesis pathways in Chinese cabbage. (**A**), Flavonoid biosynthesis pathways include DEGs and DAMs in Chinese cabbage. Green rectangles represent down-regulated DAMs, red rectangles represent up-regulated DAMs, red ovals represent up-regulated DEGs, and the bars in the figure represent the abundance of DAMs in the metabolome. (**B**), Heat map analysis of DEGs associated with flavonoid biosynthesis in RNA-seq. (**C**), The correlation matrix of DEGs and DAMs in flavonoid biosynthesis, in which orange is positive and blue is negative. The darker the color, the greater the correlation; the larger the circle, the greater the correlation. (**D**), The association networks diagram between DEGs and DAMs of flavonoid biosynthesis. The red lines represent a positive correlation, and the green lines represent a negative correlation.

**Figure 6 ijms-25-11459-f006:**
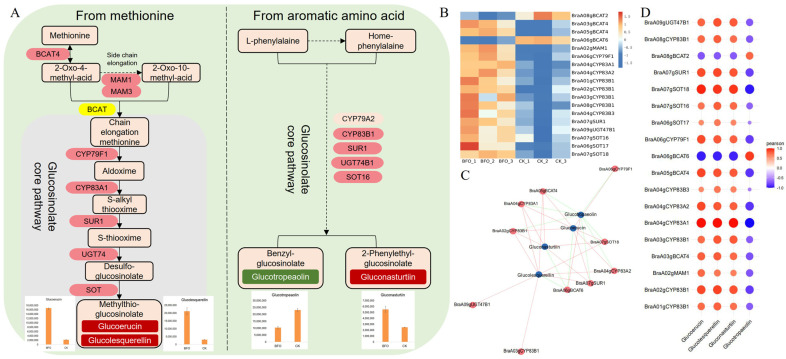
The DEGs and DAMs are involved in the pathways of glucosinolate biosynthesis in Chinese cabbage. (**A**), Glucosinolate biosynthesis pathways include DEGs and DAMs in Chinese cabbage. Green rectangles represent down-regulated DAMs, red rectangles represent up-regulated DAMs, red ovals represent up-regulated DEGs, yellow ovals represent both up-regulated and down-regulated DEGs, and the bars in the figure represent the abundance of DAMs in the metabolome. (**B**), Heat map analysis of DEGs associated with glucosinolate biosynthesis in RNA-seq. (**C**), The association networks diagram between DEGs and DAMs of glucosinolate biosynthesis. The red lines represent a positive correlation, and the green lines represent a negative correlation. (**D**), The correlation matrix of DEGs and DAMs in glucosinolate biosynthesis, in which orange is positive and blue is negative. The darker the color, the greater the correlation; the larger the circle, the greater the correlation.

**Figure 7 ijms-25-11459-f007:**
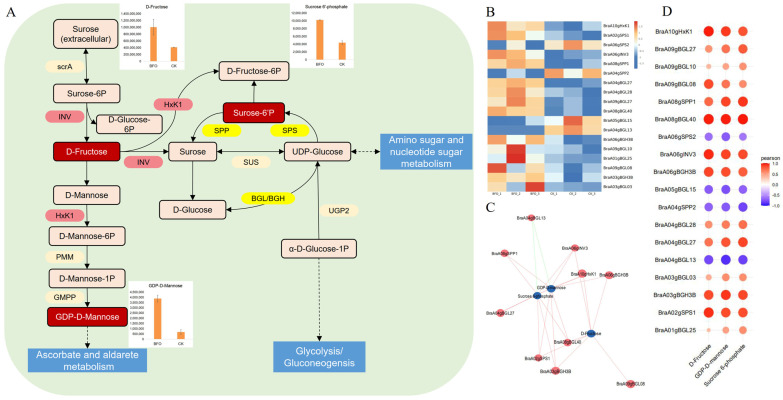
The DEGs and DAMs are involved in the pathways of soluble sugar metabolism in Chinese cabbage. (**A**), Soluble sugar metabolism pathways include DEGs and DAMs in Chinese cabbage. Green rectangles represent down-regulated DAMs, red rectangles represent up-regulated DAMs, red ovals represent up-regulated DEGs, yellow ovals represent both up-regulated and down-regulated DEGs, and the bars in the figure represent the abundance of DAMs in the metabolome. (**B**), Heat map analysis of DEGs associated with soluble sugar metabolism in RNA-seq. (**C**), The association networks diagram between DEGs and DAMs of soluble sugar biosynthesis. The red lines represent a positive correlation, and the green lines represent a negative correlation. (**D**), The correlation matrix of DEGs and DAMs in soluble sugar biosynthesis, in which orange is positive and blue is negative. The darker the color, the greater the correlation, and the larger the circle, the greater the correlation.

**Figure 8 ijms-25-11459-f008:**
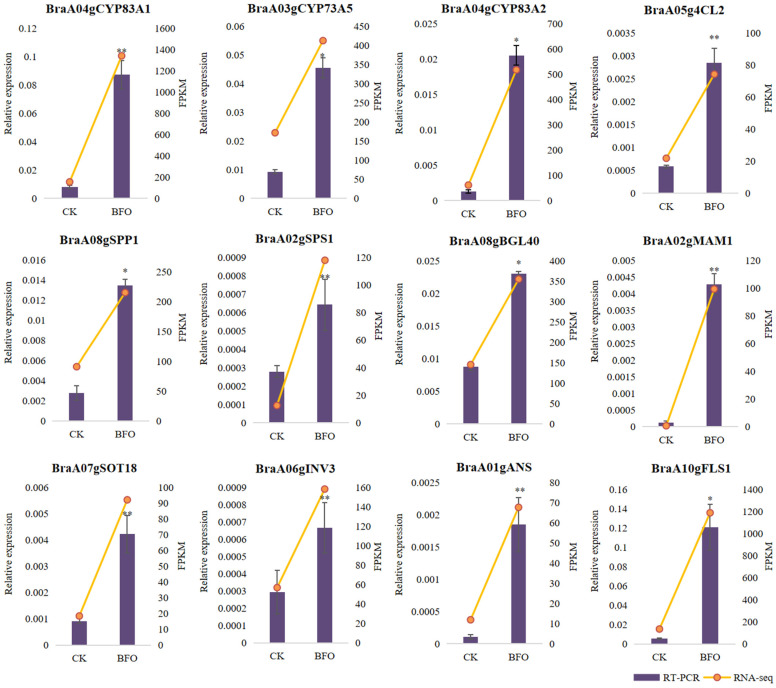
Identification of DEGs in RNA-seq by qRT-PCR. After 10 g·L^−1^ BFO treatment for 3 days, the leaves were sampled with H_2_O treatment as the control (CK). The yellow broken line represents the FPKM of RNA-seq; the purple histogram represents the relative mRNA expression determined by qRT-PCR. Data are the means of three replicates (±SDs). Asterisks indicate statistical significance using Student’s *t*-test: * *p* ≤ 0.05, ** *p* ≤ 0.01.

**Table 1 ijms-25-11459-t001:** The data quality of the RNA-seq sample.

Sample	Raw_Reads	Raw_Bases	Clean_Reads	Clean_Bases	Error_Rate	Q20	Q30	GC_Pct	Total_Map
CK_1	49,091,094	7.36 G	46,527,448	6.98 G	0.01	98.81	96.35	47.51	40,042,254 (86.06%)
CK_2	48,287,128	7.24 G	45,067,680	6.76 G	0.01	98.95	96.82	47.77	38,292,249 (84.97%)
CK_3	49,014,792	7.35 G	45,731,388	6.86 G	0.01	98.89	96.66	47.69	39,031,728 (85.35%)
BFO_1	53,851,060	8.08 G	53,745,642	8.06 G	0.01	98.78	96.4	47.42	45,525,309 (84.71%)
BFO_2	50,004,696	7.5 G	45,934,370	6.89 G	0.01	98.9	96.71	47.23	39,156,919 (85.25%)
BFO_3	47,101,456	7.07 G	47,042,114	7.06 G	0.01	99.03	97.02	47.46	40,536,026 (86.17%)

Sample: sample name. raw_reads: Number of reads in the raw data. raw_bases: Number of bases in the raw data = raw reads × 150 bp. clean_reads: Number of reads filtered from the original data. clean_bases: Number of bases filtered from the raw data = clean reads × 150 bp. error_rate: Data overall sequencing error rate. Q20: The percentage of bases with Phred values greater than 20 in total bases. Q30: The percentage of bases with Phred values greater than 30 in total bases. GC_pct: The percentage of G and C in four bases in clean reads. Total_map: Compare the number and percentage of reads on the genome.

**Table 2 ijms-25-11459-t002:** The primer sequence.

Gene ID	Prime Name	Prime Sequence (5′-3′)
BraA01g013470.3C	BraA01gANS-F	TTGAAAGAGTTGAGAGCTT
	BraA01gANS-R	TTGTGGACCGTCTTCTTT
BraA02g008800.3C	BraA02gSPS1-F	ATTCTGATACCGGTGGCC
	BraA02gSPS1-R	TCTCGTCCGAGAGGTCTT
BraA02g042670.3C	BraA02gMAM1-F	ATGGTTGTCCGGTCATTC
	BraA02gMAM1-R	TGGGAGCTTGTTCGGAAT
BraA03g016250.3C	BraA03gCYP73A5-F	ATGGACCTTCTCTTGTTG
	BraA03gCYP73A5-R	GGATTGGTATAGGACCAG
BraA04g008320.3C	BraA04gCYP83A2-F	ATGGAAGATATCATCATCGG
	BraA04gCYP83A2-R	TTAACCTGGCTAAGCTG
BraA04g029510.3C	BraA04gCYP83A1-F	AAGATGTCATCATCGGC
	BraA04gCYP83A1-R	GTTGTGGGTTAAGGTTCT
BraA05g025850.3C	BraA05g4CL2-F	ATGTCCACACGAGAAGAG
	BraA05g4CL2-R	CGTACTCGGAGATGTTT
BraA06g000740.3C	BraA06gINV3-F	AAGATCTCAACCAACCGT
	BraA06gINV3-R	CCCACACGATTCTTACAT
BraA07g038050.3C	BraA07gSOT18-F	ATCAGAGCCCTTAACCGT
	BraA07gSOT18-R	GTGACCACCGTACTCGAT
BraA08g002880.3C	BraA08gSPP1-F	GTGGGAACAAACTGGGAA
	BraA08gSPP1-R	GATGATCCAGTTGCTGTC
BraA08g025770.3C	BraA08gBGL40-F	ATGGTGACGATGGATAAGA
	BraA08gBGL40-R	CCATATGGTAGGACCTCT
BraA10g030950.3C	BraA10gFLS1-F	ATCGAGAGAGTCCAAGAC
	BraA10gFLS1-R	ACCTCGGAAAGTGGTGA

## Data Availability

The transcriptome raw data involved in this publication has been uploaded to NCBI’s Sequence Read Archive (SRA), and the accession numbers and website are available as follows: PRJNA1163725, https://www.ncbi.nlm.nih.gov/sra/PRJNA1163725 (accessed on 18 September 2024).

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
