# Peer review of "Integrated Analysis of Metabolome and Transcriptome Reveals the Effect of Burdock Fructooligosaccharide on the Quality of Chinese Cabbage (Brassica rapa L. ssp. Pekinensis)"

_ijms, 2024, doi:10.3390/ijms252111459_

Round 1
Reviewer 1 Report
Comments and Suggestions for Authors
The study focused on a single plant species, Chinese cabbage. Although the results are positive for this species, they may not be generalizable to other vegetable crops. Without further testing on other plants, it is difficult to assess how effective BFO is in other agricultural contexts.
The metabolome and transcriptome analyzes used in the study are advanced techniques that require specialized equipment and expertise. These methods may be difficult to replicate or use on a large scale by farmers or under commercial conditions, limiting the immediate applicability of the conclusions.
The study focuses on the short-term effects of BFO application (3 days after treatment). There is no information on long-term effects on the growth or quality of Chinese cabbage, which may limit the full understanding of the impact of BFO on the complete life cycle of the plant.
The study focuses exclusively on BFO, without providing detailed comparisons with other types of oligosaccharides or bioactive fertilizers already used in agriculture. Such a comparison would be helpful to understand the advantages and disadvantages of BFO relative to other available products.
The research shows that applying a BFO concentration higher than 10 g/L did not result in significant additional improvements. This raises questions about the economic potential of using BFO, as higher concentrations do not appear to provide commensurate benefits
the discussion part should also capture these aspects... and the conclusions part should be improved by adding new phrases
Reviewer 2 Report
Comments and Suggestions for Authors
Dear Authors,
Reviewer comments ijms-3260054
The manuscript entitled „Integrated analysis of metabolome and transcriptome reveals the effect of burdock fructooligosaccharide on the quality of Chinese cabbage (Brassica rapa L. ssp. Pekinensis)“ represents a useful study aimed at an investigation of the effect of exogenous application of burdock fructooligosaccharide (BFO) which is fructose with a low polymerization degree on Chinese cabbage transcriptome and metabolome, namely the key transcripts and metabolites associated with flavonoid and glucosinolate biosynthesis thus affecting Chinese cabbage quality. The transcriptomic RNAseq analysis was validated by qRT-PCR. I can recommend the manuscript for publication in IJMS.
However, I have some important comments on the present manuscript.
1/ Terminology:
I would recommend the authors to replace the term „differentially expressed metabolites (DEMs)“ with „differentially accumulated metabolites (DAMs)“ since the metabolites levels determined by metabolomic analysis always represent a result of both metabolite biosynthesis and its degradation.
Discussion, page 12, line 15: Correct the term „chalones“ to „chalcones“.
2/ Terminology: be consistent and use the abbreviation „DEGs“ for „differentially expressed genes“ in the whole manuscript. Do not use the abbreviation „DGEs“ (Figure 8 legend, line 1) or any other.
2/ Statistical evaluation:
In Figure 1 legend, the statistical evaluation used in all parts of Figure 1 has to be explained in the figure legend, e.g., different letters indicate significant differences as determined by ANOVA followed by Duncan´s multiple range test??
In Figure 8 legend, please remove the last statement on „Different letters indicate the significant differences between samples according to the Duncan´s new multiple range test…“ – this statement is not true since there are only two samples compared, one from RNAseq analysis and the other from qRT-PCR analysis, and the two samples can be compared by using Student T-test but not by Duncan´s multiple range test. The statement should be either removed or modified to Student T-test and the significant differences between RNAseq data and qRT-PCR data should be indicated by an asterisk.
3/ Abbreviations list: I think that a separate abbreviations list should be added to the manuscript since they can be unfamiliar to the readers.
4/ In Materials and methods, the term „growth potential seedlings“ has to be explained. Should it mean „seedling vigor“??
5/ In Materials and methods, part 5. Metabolome analysis, the frequency of ultrasonication has to eb specified, and, moreover, the centrifugation should be rather expressed as „g“ units than „rpm“ since „rpm“ depends on centrifuge type. Centrifuge type and manufacturer have to be given.
6/ In Materials and methods, part 6. Transcriptome analysis, the reference genome of Chinese cabbage has to be specified by providing the web address and date of access.
7/ Formal comments on the text related to English language and style:
Abstract, line 1: Replace „a“ from „fructose“ to „low polymerization degree“ in the statement: „Burdock fructooligosaccharide (BFO) is fructose with a low polymerization degree which could improve…“
Figure 8 legend, line 3: Correct the typing error in the word „purple“ (not „purpel“) in the statement: „…the purple histogram represents the relativem RNA expression determined…“
Results, page 9, line 1: Add a space between the word „monooxygenase“ and the abbreviations „CYP79F1, CYP83A1“.
Discussion, page 11, line 13: Replace the word „of“ with „on“ in the statement: „Therefore, 10 g L-1 BFO was the optimal concentration for exogenous application on Chinese cabbage seedlings.“
Discussion, page 12, line 9: Correct the abbreviation „DGEs“ to „DEGs“ for „differentially expressed genes“.
Discussion, page 12, line 15: Correct the term „chalones“ to „chalcones“.
Discussion, page 12, line 25: Insert a space between the words „by“ and „CHI“.
Discussion, page 13, line 10: The gene name „SOT“ should be explained in the abbreviations list.
Discussion, page 13, line 27: Modify the verb form „increased“ to „increasing“ in the statement: „…and GDP-D-mannose accumulated under BFO treatment significantly increasing the soluble sugar content of Chinese cabbage (Figure 7A).“
Discussion, page 13, line 30. Modify the verb „plays“ to „which play“ in the statement: „Moreover, GDP-D-mannose is a molecule that serves as a source of mannosyl units in various glycoconjugates which play a crucial role in biochemical processes….“
Discussion, page 13, line 35: Use the full enzyme name for the first time and add the abbreviation into abbreviations list in „Invertase (INV) cleaves cytoplasmic sucrose to produce D-fructose and D-glucose-6P,…“
Final recommendation: Accept after a minor revision.
